# Social Capital and the Realization of Mutual Assistance for the Elderly in Rural Areas—Based on the Intermediary Role of Psychological Capital

**DOI:** 10.3390/ijerph20010415

**Published:** 2022-12-27

**Authors:** Xinglong Xu, Lingqing Zhao

**Affiliations:** School of Management, Jiangsu University, 301 Xuefu Road, Zhenjiang 212013, China

**Keywords:** social capital, psychological capital, rural elderly, mutual assistance, intermediary effect

## Abstract

Background: Mutual assistance for the elderly is a new pension model that has been widely valued and discussed in China, especially in rural areas. The social and psychological capital owned by the elderly in rural areas promotes their participation and affects the realization of mutual assistance for the elderly. Based on this, this paper proposes the following hypotheses: H1: Bonding social capital positively affects the realization of mutual assistance for the elderly in rural areas; H2: bridging social capital positively affects the realization of mutual assistance for the elderly in rural areas; H3: linking social capital positively affects the realization of mutual assistance for the elderly in rural areas; and H4: psychological capital plays an intermediary role in the process of social capital influencing the realization of mutual assistance for the elderly in rural areas. The empirical research is carried out around these hypotheses. Methods: In this paper, the 2019 China General Survey of Social Conditions (CSS) database was used as the data source for empirical analysis. Social capital can be divided into bonding social capital, bridging social capital, and linking social capital, while psychological capital can be divided into four dimensions: self-efficacy, optimism, hope, and resilience. Our evaluation was performed using LOGIT regression analysis with STATA16 software. First, the correlation of social capital to the realization of mutual assistance for the elderly in rural areas was verified. Next, the mediation effect was verified using the KHB regression method, and the influence of psychological capital as an intermediary variable on the realization of mutual assistance for the elderly in rural areas was demonstrated. Results: Social capital had a significant positive effect on mutual assistance for the elderly in rural areas. Psychological capital played an intermediary role in the relationship between the three types of social capital and rural mutual assistance for the elderly. Among the four dimensions of psychological capital, self-efficacy, optimism, and tenacity played a partial mediating role in the relationship between social capital and mutual assistance for the elderly in rural areas, while the mediating role of hope was not significant. Conclusions: (1) All three types of social capital have a significant promoting effect on the realization of mutual assistance for the elderly in rural areas, among which bridging social capital has the most significant effect. (2) Psychological capital plays a partial mediating role in the three kinds of social capital’s influence on the realization of mutual assistance for the elderly in rural areas, and the intermediary role is the strongest in the effect of linking social capital on the realization of mutual assistance for the elderly in rural areas, but the overall effect is not high. (3) Among the four dimensions of psychological capital, self-efficacy, optimism, and tenacity all have certain intermediary effects, but the intermediary effect of hope is not significant. (4) There are significant gender and regional differences in the impact of social capital and psychological capital on the realization of mutual assistance for the elderly in rural areas.

## 1. Introduction

Family endowment and individual self-endowment are effective measures to implement the national strategy of actively dealing with population aging in China. At this stage, however, the family endowment function is gradually weakening [1]. In rural areas, there is a large outflow of young children, leaving the “nest empty” and increasing the left-behind elderly proportion, which weakens child care and the emotional support for parents. This affected the rural elderly’s physical and mental health [2]. Influenced by traditional concepts, some of the elderly in rural China live a simple life, have a modest diet, ignore health, and are unwilling to add a pension burden to their children, which leads to low efficiency of their self-pension function. In this context, both the Chinese government and academic circles strongly advocate mutual assistance for old-age care [3] and regard it as an important task of aging work in the new era. In 2021, the Opinions of the CPC Central Committee and the State Council on Strengthening the Work on Aging in the New Era proposed to “encourage the development of mutual-assistance elderly care services based on the village level neighborhood mutual assistance points and rural happiness homes in combination with the implementation of the rural revitalization strategy,” showing the important role of rural mutual assistance for elderly care in China’s rural revitalization and the work on aging in the new era. So, how should the mutual assistance for the elderly in rural areas be realized? This paper will study this problem.

Mutual assistance for the elderly is under government support and guidance, relying on the social organization. Villagers’ autonomous organizations, relatives and neighbors, the ordinary village, and volunteer support networks, especially (quasi) older human resources effectively organized as a service power, provide low-cost endowment services geared to the needs of the rural elderly and belong to the typical form of pension service cooperative production. Its soul lies in mutual assistance [4].

The active participation of the rural elderly is considered the primary premise of realizing mutual assistance for the rural elderly. In the current research on promoting the participation of the rural elderly in mutual assistance for the elderly, social capital is generally regarded as an important influencing factor [5]. However, the concept of social capital depends on the context, but most of the studies on China’s rural mutual assistance for the elderly lack the division of social capital in the context of rural mutual assistance for the elderly. At the same time, although the synergistic effect of psychological capital and social capital has been studied and discussed more in other fields of social capital research, psychological capital is rarely applied in the relevant research on the realization of rural mutual assistance for the elderly in China. Therefore, the purpose of this study is to examine the direct effects of social capital on the realization of mutual assistance for the elderly in rural China and the intermediary role of psychological capital so as to explore the promotion factors for the rural elderly to participate in mutual assistance for the elderly, help local governments determine the focus of policy intervention, and promote the realization of mutual assistance for the elderly in rural areas.

### 1.1. Defining Social Capital

The concept of social capital was first formally proposed by Bourdieu [6], and later Coleman developed the theory to the meso level [5]. Robert D. Putnam popularized it on a large scale and enriched the connotation to include three dimensions: network, trust, and norm [7]. Lin’s research has its distinctive characteristics: it emphasizes the role of individuals in the formation and use of social capital [8]. The available research scope of social capital is wide. In existing research, it is usually one of the strategies to study the problems of old age from the perspective of social capital [9,10,11].

In China, social capital is a relatively stable, institutionalized, and sustainable social relationship network, which usually exists in the neighborhood relationship, professional relationship, kinship relationship, and organizational relationship. In fact, there are two main reasons to study social capital in China. First, due to the current objective differences between urban and rural areas, rural areas have far fewer formal institutions and resources than urban areas. In this context, social capital maintained based on kinship, geographical, and industrial relationships is particularly important for rural families’ behavior choice. Second, in China, especially in rural China, a typical nepotist society, the role of social capital in changing, and promoting residents’ behavior has always been valued. Therefore, social capital is often studied as a reliable solution to the problems associated with rural China. Some scholars believe that social capital plays an important role in promoting Chinese farmers’ health [12] and family division of labor [13]. However, other scholars point out that social capital does not always have good effects [14]. 

At the same time, social capital has strong situational characteristics. These characteristics determine that social capital should be classified according to its application field. In the current research on rural social capital, there are many ways to divide social capital into different categories. Yue and Liang [15] classified the social networks of the rural elderly based on relationships, such as relatives and neighbors, as structural social capital and classified the trust, mutual assistance, and reciprocity of the rural elderly as cognitive social capital. Based on China’s social characteristics, Lu et al. further distinguished structural social capital and cognitive social capital from different sources of social capital at the family and community levels to study the relationship between the structural social capital and self-rated health of the rural elderly [16]. Nie classified the kinship relationship of the rural elderly in the context of mutual assistance for the elderly as a “strong relationship” network and the social relationships, such as neighbors, friends, and villagers, as a “weak relationship” network [17]. According to the division method proposed by Putnum (2000) [7] and Woolcock (2001) [18], Xie and Yao [19] divided farmers’ social capital into three types, compact, linking, and bridging from traditional, horizontal, and vertical perspectives. On the basis of these classification methods, combined with the related policy support of rural mutual assistance for the elderly in China, this paper divides the social capital owned by the rural elderly into bonding social capital, bridging social capital, and connecting social capital. The specific explanations are as follows: (1) As age increases, the social scope of the elderly gradually shrinks, only including close relatives, close neighbors, and individual friends who have known each other for many years. This kind of solid relationship network between relatives and friends itself has a stronger influence on people, so this group is more likely to drive the participation of rural elderly people in mutual assistance for old-age care. At the same time, the shrinking social scope inevitably leads to the intensification of the degree of information closure, especially when the rural elderly are generally not good at using modern information technology means, such a social network composed of close relatives and close neighbors, and relatives and friends may become the main channel for the rural elderly to obtain mutual assistance for the elderly so as to further strengthen its effect on the realization of mutual assistance for the elderly in rural areas. This paper defines the strong relationship network between relatives and friends of the elderly in rural areas as associative social capital. (2) Under the realistic background of “aging” and “hollowing out” in rural China, the emotional interaction between the elderly and their offspring has reduced, and some close neighbors and old friends follow their children to the cities, resulting in the loss of inherent associative social capital and emotional loss for the rural elderly. In this context, the elderly in rural areas may expand their social network by developing some hobbies or occupational activities so as to obtain resources for mutual assistance for old-age care. Therefore, this paper defines the social network relationships that the rural elderly get acquainted with based on interests, beliefs, or work as bridging social capital. (3) Mutual endowment is also a key measure to cope with the aging of the population in China. Local governments have been setting up “rural happiness homes,” “care centers for the elderly,” and other mutual assistance institutions for the elderly and, through various channels, guide and encourage the rural elderly to participate in mutual assistance; therefore, in this paper, the social capital acquired by the rural elderly through policy support is defined as linking social capital.

### 1.2. Bonding Social Capital and the Realization of Mutual Assistance for the Elderly in Rural Areas

Bonding social capital is mainly embodied in the sharing of resources and trust and reciprocity among relatives, friends, and neighbors. Liu [20] pointed out that friends, relatives, and neighbors help the elderly to obtain social resources for mutual assistance so as to realize mutual assistance for the elderly. Wang showed that pension resources among relatives and friends could significantly reduce the pressure of basic life security and family care for the elderly and enable them to participate in social communication activities and engage in mutual assistance for the elderly [21]. On the intangible level, trust is also the main characteristic of associative social capital. At present, mutual assistance for old-age care is still in the early stage of development, and relevant rules and regulations need to be improved [22]. It is inevitable that the elderly will worry about the implementation of old-age care services in the process of intergenerational exchange and have doubts about the new old-age care model. Long-term mutual trust with friends, relatives, and neighbors can improve the fulfilment expectation of elderly care services and eliminate concerns caused by information barriers. Compared with other types of trust, the trust between neighbors and friends has a stronger promoting effect on the willingness of the rural elderly to help each other in old age [23]. Liu believes that the low trust of the rural elderly, coupled with their mentality of catering to government directives and shifting responsibilities, is an important reason for the difficulty in forming mutual assistance and cooperation [24]. It is not difficult to find that the richer the bonding social capital is, the more material and spiritual support the rural elderly can obtain, and the rural elderly with more profits are more likely to participate in mutual assistance for the elderly.

### 1.3. Bridging Social Capital and the Realization of Mutual Assistance for the Elderly in Rural Areas

Bridging social capital is mainly embodied in the “weak relationship” social network established between the rural elderly and unfamiliar groups. In addition, bridging social capital can promote the social cognition of the elderly. Adler and Kwon pointed out that bridging social capital has more advantages of value information resources than the social capital transmitted by close internal networks [25]. Nie found that communication with members with “weak ties” helps the elderly obtain heterogeneous and less redundant information, thus promoting their resource exchange and reciprocal behavior [17]. Mi and Li believe that as a new mode of rural old-age care, the comprehensive and objective cognition of the participants is the basis for its effective operation [26]. The research of Xu also confirmed that the conceptual elements of mutual care for the elderly are the key for the elderly to give positive meaning to the participation behavior of mutual care for services for the elderly and make participation choices [27]. It is not difficult to see that rich bridging social capital helps the rural elderly have a wide range of information access channels and strengthen their awareness of social mutual endowment resources so as to obtain more effective mutual endowment security conditions. 

### 1.4. Linking Social Capital and the Realization of Mutual Assistance for the Elderly in Rural Areas

Linking social capital is mainly reflected in policy support. It refers to all kinds of spiritual or material support or help obtained by individuals from corresponding organizations or departments. In essence, it is a kind of support for individuals’ material, life, and psychology, which is consistent with the current situation that the government encourages and supports the elderly to voluntarily participate in mutual assistance for the elderly through policies. Scholars mostly divide this kind of social support into instrumental support, emotional support, and information support [28]. Instrumental support is embodied in the activities that governments at all levels carry out for the elderly in the community. The more frequent the activities for the elderly, the higher the community care, and the higher the physical and mental health of the elderly and the trust level of the government so as to mobilize the enthusiasm of the elderly to help each other [29]. Emotional support is embodied in state-built rural happiness centers, day activity centers for the elderly, and other mutual assistance centers for the elderly. Such places can be used by the elderly with low or free compensation and become the main resources affecting the mental health of the rural elderly [30]. Information support is reflected in the government’s publicity of the connotation, operation mode, and corresponding benefits of mutual assistance for the elderly through various news media, which also plays a certain role in promoting the realization of mutual assistance for the elderly in rural areas [31]. 

### 1.5. The Mediating Role of Psychological Capital

In the existing studies on the relationship between social capital and mutual care for the elderly, most scholars adopt the “stimulus response” research paradigm, in which the elderly seem to accept endowment resources and mechanically transform them into pension benefits to produce mutual care for the elderly. In fact, social capital is not completely equivalent to inherent resources. Even if different individuals have the same social capital, it only means that they have equal opportunities to use that social capital but not the same actual use efficiency. The final transformation effect is also affected by psychological capital, such as individual motivation, cognition, and personality [32]. Namkung [33] and other scholars have pointed out that in the process of external factors influencing individual behavior, the individual’s mental state and a series of psychological reactions play a role.

The psychological capital of the rural elderly mainly includes four dimensions of self-efficacy, hope, optimism, and tenacity, which is considered an important way to transform individual potential into practical ability [34]. Specifically, “self-efficacy” refers to an individual’s confidence in mobilizing and using resources to successfully complete a specific task and goal in order to achieve motivation [35], which is believed to be conducive to improving the elderly’s confidence in life, improving their quality of life, and promoting active aging [36] and then promoting rural elderly mutual care. In terms of hope, Luthans [37] believes that it is the primary condition for creating value. Individuals with high hope levels take the initiative to seek different methods and approaches to achieving their goals. For the rural elderly, hope means a greater inclination to try various types of old-age care methods in order to realize “providing for old age” and “enjoying old age.” Optimism refers to the individual’s expectation of positive effects, which can help the rural elderly identify high-quality endowment resources, pursue goals, and deal with difficulties. Therefore, the optimistic elderly are more likely to choose mutual assistance for the elderly. Individuals with high tenacity can show higher adaptability in the face of negative influences and changing environments [38], which is also conducive to the participation of the rural elderly in mutual assistance for the elderly. Generally speaking, the elderly in rural areas have a relatively rich endowment and ability of psychological resources, which means they have the courage and confidence to develop better in future life and thus improve the probability of participating in mutual assistance for the elderly.

To sum up, rural revitalization is of great strategic significance in China. In the current situation of aging in China’s rural areas, it has become an important part of realizing rural revitalization to properly solve the problem of old-age care for the rural elderly. As a new way to solve the aforementioned problems, it is of great theoretical and practical significance to discuss and promote the realization of rural mutual assistance for the elderly. From the perspective of the synergy of social capital and psychological capital, this paper provides a promotion strategy for the realization of mutual assistance for the elderly in rural areas and helps local governments take targeted measures. In contrast, from the perspective of social capital research, given the differences in socio-economic status and culture, it is important to conduct social capital research in the environment of developing and poor communities. Previous research shows that social capital may have a greater impact on the well-being of the rural elderly with lower socio-economic status than the wealthier elderly. In addition, although increasingly more scholars are paying attention to the impact of social capital and psychological capital on the realization of mutual assistance for the elderly in rural areas, there is little research on the synergy of these three elements. To be specific, few studies have systematically proved the intermediary role of psychological capital in the relationship between social capital and the realization of rural mutual assistance for the elderly. Therefore, based on the rural background of China, this paper studies the role of social capital in promoting the realization of mutual assistance for the elderly in rural areas and the intermediary role of psychological capital in it, which will also help develop and deepen the current research on social capital and expand the scope of application of psychological capital.

Accordingly, this paper proposes the following hypotheses:

**H1.** 
*Bonding social capital positively affects the realization of mutual assistance for the elderly in rural areas.*


**H2.** 
*Bridging social capital positively affects the realization of mutual assistance for the elderly in rural areas.*


**H3.** 
*Linking social capital positively affects the realization of mutual assistance for the elderly in rural areas.*


**H4.** 
*Psychological capital plays an intermediary role in the process of social capital influencing the realization of mutual assistance for the elderly in rural areas.*


**H4a.** 
*Self-efficacy plays a mediating role in the process of social capital influencing the realization of mutual assistance for the elderly in rural areas.*


**H4b.** 
*Optimism plays a mediating role in the process of social capital influencing the realization of mutual assistance for the elderly in rural areas.*


**H4c.** 
*Hope plays a mediating role in the process of social capital influencing the realization of mutual assistance for the elderly in rural areas.*


**H4d.** 
*Tenacity plays a mediating role in the process of social capital influencing the realization of mutual assistance for the elderly in rural areas.*


## 2. Research Design and Method Section

### 2.1. Data Source and Processing

The data used in this paper are from the 2019 China General Social Survey (CSS). The Chinese Academy of Social Sciences Institute of Sociology conducted a 2-year longitudinal, comprehensive survey of Chinese society. The survey area covered the 31 national provinces/autonomous regions/municipalities directly under the central government, including 151 cities and counties and 604 villages/residents’ committees. The selected 2019 panel data contained a sample of 10,283 urban and rural residents of all ages. Because the research mainly focuses on the realization of mutual assistance for the elderly in rural areas, the research subjects were limited to residents over 60 years old with rural household registration. Therefore, the samples selected in this paper met the conditions of both agricultural household registration and birth before 1962, and a total of 1708 valid sample data were obtained. At the same time, the CSS database has pre-divided all samples into three regions, east, middle, and west, which is convenient for further classification of sample regions. The specific sample numbers and regional distribution are shown in Table 1.

### 2.2. Variable Interpretation

**(i) Dependent variable.** The dependent variable was the realization of rural elderly mutual assistance. Some scholars subdivide rural mutual assistance for the elderly into spiritual comfort, cultural entertainment, daily life, medical examination and rehabilitation, and other items [39]. Therefore, based on the existing research and the basic connotation of mutual assistance for the elderly, and combined with the design of the 2019 Comprehensive Survey of Social Conditions in China (CSS) questionnaire, this paper selected the rural elderly to participate in four types of voluntary services, namely elderly care, medical care, helping the disabled, and helping the poor, to measure mutual assistance for the rural elderly. See Table 2 for specific scale designs.

**(ii) Independent variable.** The independent variable was social capital. This paper divided social capital into three categories, bonding type, bridging type, and linking type, and studied its influence on the realization of mutual assistance for the elderly in rural areas. Due to the inconsistency in the transformation and identification ability of different individual resources, the potential available social capital in the social network of the rural elderly cannot be equated with the actual amount of social capital. Therefore, based on the perspective of the availability of social capital [40], the bonding, bridging, and linking social capital of the rural elderly is measured from four aspects: social network scale, social network intensity, social network environment, and social information exchange. See Table 2 for specific scale designs.

**(iii) Mediation variable.** The mediating variable was the psychological capital of the rural elderly. According to the psychological capital scale of Luthans et al. [41], this paper chose four dimensions of self-efficacy, optimism, hope, and tenacity to explain the connotation of psychological capital and measured psychological capital in combination with the items included in CSS 2019. See Table 2 for specific scale designs.

**(iv) Control variables.** The control variables were mainly the basic population variables. Following the practice of the existing authoritative literature and combined with the availability of data, the control variables in this paper were divided into three levels: individual characteristics, family characteristics, and regional characteristics. Individual characteristics included age, sex, religion, degree of education, and marital status; family characteristics included the relative economic status, family size, and number of children; at the regional level, the east, central, and western dummy variables were added. The setting and assignment of control variables are shown in Table 2.

## 3. Methods Section and Model Construction

### 3.1. Baseline Regression Model

To test the impact of bonding social capital, bridging social capital, and linking social capital on psychological capital and the realization of mutual assistance for the elderly in rural areas, this paper constructed the following benchmark regression model:(1)$$Y\;={\;B}_{0}+B_{11}X_{1}+B_{12}X_{2}+B_{13}X_{3}+B_{K}C+\varepsilon$$
(2)$$Z={\;B}_{1}+B_{21}X_{1}+B_{22}X_{2}+B_{23}X_{3}+B_{K}C+\varepsilon \;$$

In Formula (1), Y is the dependent variable (the realization of mutual assistance for the elderly in rural areas) and B_0_ is a constant term. B_11_, B_12_, and B_13_, respectively, refer to the influence of bonding social capital, bridging social capital, and linking social capital on the realization of mutual assistance for the elderly in rural areas when other variables are controlled. In Formula (2), Z is the intermediary variable (psychological capital of the rural elderly) and B_1_ is a constant term. B_21_, B_22_, and B_23_, respectively, refer to the influence of bonding social capital, bridging social capital, and linking social capital on the psychological capital of the rural elderly. C represents the control variable of this study, B_K_ is the coefficient of influence of the control variable on the realization of mutual assistance for the elderly in rural areas, and ε is a random error term.

### 3.2. The Mediation Decomposition Effect of KHB

In this paper, the KHB method [42] was used to estimate the mediating variables of nonlinear models and a regression model of mediating effects was constructed. The KHB analysis method is suitable for estimating the intermediate variables of nonlinear probability models, including the binary logit regression model, and the mediating effect can be decomposed by setting the relationship among the dependent, initial independent, and mediating variables. Specifically, it involves three steps:

Step 1: Set up a simplified model to test the total effect of different types of social capital on the realization of mutual assistance for the elderly in rural areas:(3)$$Y\;={\;\alpha }_{1}+\beta_{1}X_{1}+\beta_{2}X_{2}+\beta_{3}X_{3}+\delta_{1}C+\varepsilon$$

Step 2: Set up the complete model to test the direct effect of different classification groups on the realization of mutual assistance for the elderly in rural areas:(4)$$Y=\alpha_{2}+\beta_{1}^{\prime }X_{1}+\beta_{2}^{\prime }X_{2}+\beta_{3}^{\prime }X_{3}+Z\gamma_{1}+C\delta_{2}+\varepsilon$$

In the third step, the difference coefficient ($\beta_{x}-\beta_{x}^{\prime }$) can be obtained from the total effect and direct effect, which is the indirect effect of the core independent variables X_1_, X_2_, and X_3_, namely the mediating role played by variable Z. If the difference coefficient is significantly positive, it indicates that the coefficient of variable X decreases under the action of the mediating variable, that is, the mediating variable has a mediating effect on it. If the difference coefficient is significantly negative, it indicates that the mediating variable Z has a restraining effect on it.

The specific measurement results were completed using STATA16 software.

## 4. Results and Discussion

### 4.1. Descriptive Statistics

To facilitate the analysis, some variables were operationalized in this paper. Among them, the scores of the dependent, independent, and mediating variables were the arithmetic average scores of each item. The results of the descriptive statistical analysis are shown in Table 3. It is not difficult to see that the average mutual assistance for the rural elderly was 1.08, which shows that the mutual assistance for the rural elderly in China is not high. The average bonding social capital was 2.14, and that of bridging social capital was 1.266, indicating that the rural elderly are relatively rich in bonding social capital stock and especially lacking in bridging social capital stock, which is also consistent with the conclusion that the social range of the rural elderly shrinks with the increase in age. It is worth noting that the difference between the minimum and maximum of bonding social capital was large, which indicates that there is a great difference in the stock of bonding social capital among the rural elderly. The average value of linking social capital was 1.423, indicating that the elderly in rural areas feel that the level of policy support is not high. The average value of psychological capital was more than 1.5, indicating that the rural elderly are relatively rich in psychological capital and have more positive psychological quality.

### 4.2. Correlation Analysis

To avoid the research error caused by the multicollinearity problem, a variance inflation factor test was carried out on the variables. The results showed that the variance inflation factor (VIF) was less than 1.5, and there was no high multicollinearity problem. Next, Pearson correlation analysis was conducted on the core variables to investigate whether there is a correlation between the variables. The results are shown in Table 4. It was found that bonding social capital, bridging social capital, and linking social capital are significantly positively correlated with rural mutual assistance for the elderly, which preliminarily verifies the hypothesis in this paper.

### 4.3. Benchmark Regression Analysis Results and Discussion

Since the dependent variable in this paper was a categorical variable, the correlation between bonding social capital, bridging social capital, and linking social capital and mutual assistance for the elderly in rural areas was tested by establishing a logit regression model. In addition, to avoid the homogeneity of social capital affecting the test results, the influence of the square of the independent variable on the dependent variable was further tested. The specific regression results of the model are shown in Table 5.

Model I was the regression model when only control variables are included. Model II added the independent variable social capital on this basis to explore the influence of social capital on dependent variables. The results showed that bonding social capital has a significant positive impact on mutual assistance for the rural elderly, indicating that the resource sharing and trust reciprocity between the rural elderly and their relatives and neighbors are conducive to the realization of mutual assistance for the rural elderly, namely hypothesis H1 is established. Bridging social capital had a significant impact on mutual assistance for the rural elderly, and bridging social capital showed the strongest significance in the process of the three types of social capital affecting the realization of mutual assistance for the rural elderly (β = 0.710, *p* ˂ 0.01), which indicates that the “weak relationship” network established with strangers has an important and significant positive impact on the realization of mutual assistance for the rural elderly. So, hypothesis H2 is established. However, it is worth noting that according to the research results of Sorensen, the stock of bonding social capital in rural areas is significantly greater than that of bridging social capital [43], and the amount of bridging social capital of the elderly in rural areas is also showing a decreasing trend [44]. Therefore, nurturing social capital for the rural elderly is of great significance for the realization of mutual assistance for the elderly. Linking social capital had a significant positive impact on the participation of the rural elderly in mutual assistance for the elderly (β = 0.441, *p* ˂ 0.01), which indicates that the policy support, including instrumental support, emotional support, and information support, can reduce the perceived risks of the rural elderly so that they have the confidence to deal with the problems caused by participation in mutual assistance for the elderly, such as leaving work and falling into poverty, providing services to others, but failing to get help from others. Therefore, they are more inclined to participate in government-advocated mutual pension activities, which means hypothesis H3 is valid. At the same time, after the square treatment of the three types of social capital, its impact on the realization of mutual assistance for the elderly in rural areas is still significant, which indicates that the conclusion of this study is robust.

Table 6 examines the effects of bonding social capital, bridging social capital, and linking social capital on the intermediary variable, psychological capital. Regression estimation results showed that all three kinds of social capital can significantly improve the psychological capital level of the rural elderly (bonding SC: β = 0.025, *p* < 0.01; bridging SC: β = 0058, *p* < 0.05; linking SC: β = 0.150, *p* < 0.01). The reason is that the bonding social capital represented by relatives and friends can provide a positive emotional experience and help the rural elderly have a good mood. Bridging social capital built by business ties and interest ties can bring a younger mentality to the rural elderly through the communication among peers. Social support and help from the government can provide basic security for the elderly so that they can maintain their hope for life, which is conducive to the accumulation of psychological capital.

### 4.4. Analysis of the Intermediary Mechanism of Psychological Capital to Realize Mutual Assistance for the Elderly in Rural Areas

To test the intermediary role of psychological capital, this study used the KHB method to test the mediating effect. As shown in Table 7, models V–VII indicated that bonding social capital, bridging social capital, and linking social capital can, respectively, influence the psychological capital of the rural elderly to promote their mutual assistance. The KHB intermediary test showed that for each independent variable alone, psychological capital explained 11.86% of the impact of bonding social capital on the realization of mutual assistance for the elderly in rural areas, 8.62% of the impact of bridging social capital on the realization of mutual assistance for the elderly in rural areas, and 14.86% of the impact of linking social capital on the realization of mutual assistance for the elderly in rural areas. This result shows that psychological capital plays a partial mediating role in the process of bonding social capital, bridging social capital, and linking social capital, affecting the realization of mutual assistance for the elderly in rural areas, which means hypothesis H4 is preliminarily valid.

To further explore the intermediary role of psychological capital in social capital and the realization of mutual assistance for the elderly in rural areas, this paper divided psychological capital into four dimensions, self-efficacy, optimism, hope, and tenacity, and tested their intermediary effects on social capital and the realization of mutual assistance for the elderly in rural areas. The test results are shown in Table 8 for models VIII–XII.

Model VIII was the mediating effect test of overall psychological capital in the realization of social capital and mutual assistance for the elderly in rural areas. The results showed that psychological capital plays a significant mediating effect (β = 0.350, *p* < 0.01), and its effect ratio is 12.06%, which is basically consistent with the previous test results on the intermediary effect of psychological capital on different dimensions of social capital and the realization of mutual support for the elderly in rural areas. Models IX–XII tested the mediating effect of different dimensions of psychological capital on the realization of social capital and mutual assistance for the elderly in rural areas after the division of psychological capital. The results of model IX showed that the self-efficacy dimension of psychological capital has a significant mediating effect on the realization of social capital and mutual assistance for the elderly in rural areas (β = 0.147, *p* < 0.01), and its effect ratio is 5.13%, which means hypothesis H4a holds. The results of model X showed that the optimism dimension of psychological capital has a significant mediating effect on the realization of social capital and mutual assistance for the elderly in rural areas (β = 0.113, *p* < 0.1), and its effect ratio is 3.91%, which means hypothesis H4b holds. The results of model XI showed that the hope dimension of psychological capital has no significant mediating effect on the realization of social capital and mutual assistance for the elderly in rural areas (β = −0.001, *p* > 0.1), which means hypothesis H4c does not hold. The possible reason is that at the present stage, there are not many practical ways of mutual assistance for the elderly in rural areas and the support is limited. Therefore, the elderly with a high level of hope may have too high expectations for mutual assistance for them, leading to their needs for pension services not being met, thus affecting their participation in the realization of mutual assistance for the elderly. The results of model XII showed that the tenacity dimension of psychological capital has a significant mediating effect on the realization of social capital and mutual assistance for the elderly in rural areas (β = 0.066, *p* < 0.05), and its effect ratio is 2.28%, which means hypothesis H4d holds. Therefore, part of hypothesis H4 holds. It is worth noting that there are some differences between the research results of scholars in the current collaborative research of social capital and psychological capital. Some scholars believe that the role of psychological capital is more significant [29,39], while others have affirmed the role of all four dimensions of psychological capital [45], which contradicts the results of this study. These results show that the dimensions of self-efficacy, optimism, and tenacity of psychological capital have a mediating effect on the realization of social capital and mutual assistance for the elderly in rural areas, but the effect is not high on the whole. The mediating effect of the hope dimension is not significant. 

### 4.5. Heterogeneity Verification of the Impact of Social Capital and Psychological Capital on the Realization of Mutual aid for the Elderly in Rural Areas

In addition, because the traditional Chinese gender differences in ideas, division of labor, and the objective differences in economic developmental levels and folk customs between different regions may affect the impact of social capital and psychological capital on the realization of mutual assistance for the elderly in rural areas, this paper divided the subjects into groups according to gender/region to test the difference in the relationship between social capital, psychological capital, and the realization of mutual assistance for the elderly in rural areas based on gender and region. The inspection results are shown in Table 9.

In Table 9, models XIII and XIV are regression results after grouping based on gender. The results showed that for males, linking SC (β = 1.906, *p* ˂ 0.01), PC (β = 2.298, *p* < 0.01) had greater effects on the mutual assistance of rural elderly than females, while for females, Bonding SC (β = 0.729, *p* < 0.05) and Bridging SC (β = 3.232, *p* < 0.01) played a more important role in influencing their participation in mutual aged care. The reason for this is that the traditional Chinese pattern of “male outside, female inside” determines that males’ attention and awareness of policy information is significantly higher than that of females [45], so it is easier to recognize and find support from the government and then choose to participate in mutual assistance for the elderly. At the same time, males in traditional families are the main labor force responsible for livelihood. The psychological pressure is greater and more unstable than that for females, so the psychological factors have a more significant impact on their participation in mutual aid pension. For females, the family is the main object of concern, so social capital from close relatives has a stronger impact on them. At the same time, in rural China, there has been a long-term social model of chatting with village females to relieve boredom, and this social model plays a strong role in information exchange for rural females. Therefore, the influence of this weak relationship network on females is also significantly greater than that on males. Models XV, XVI, and XVII are regression results grouped based on different regions. The results showed that bonding SC has significantly promoted the realization of mutual assistance for the elderly in rural areas only in the western region (β = 1.286, *p* ˂ 0.01), while bridging SC plays a positive role in promoting the participation of the rural elderly in mutual aid pension in the whole region (eastern region β = 1.755, *p* ˂ 0.05; central region β = 3.118, *p* ˂ 0.01; western region β = 1.083, *p* ˂ 0.1). Linking SC significantly promotes the realization of mutual assistance for the elderly in the rural areas of only central China (β = 1.563, *p* ˂ 0.1). This research result is consistent with the previous conclusion that bridging SC plays the most significant role in promoting the realization of mutual assistance for the elderly in rural areas. In addition, psychological capital plays a more significant role in promoting the realization of mutual assistance for the elderly in eastern and western regions (eastern region β = 2.321, *p* ˂ 0.01; western region β = 2.868, *p* ˂ 0.01). According to the analysis, although the economic developmental levels of the eastern and western regions are quite different, according to the hierarchy of demand theory, the elderly in the eastern region may choose to participate in mutual aid pension based on self-realization or social needs, while the elderly in the western region are more inclined to obtain security through participating in mutual aid pension, so a healthier psychological level will help them choose to participate in mutual aid pension; the effect of individual psychological capital is not significant in the central region, maybe because of the existence of a mutual aid culture for a long time.

## 5. Conclusions

Mutual assistance for the elderly is an effective tool to deal with the shortage of rural elderly service supply, and the effective participation of the rural elderly is the basis of its realization. This paper divided the social capital of the rural elderly in the context of mutual assistance into bonding social capital, bridging social capital, and linking social capital; constructed a theoretical framework of social capital’s influence on the realization of mutual assistance for the rural elderly; and tested the mediating role of psychological capital in the process of social capital’s influence on the realization of mutual assistance for the rural elderly. The results are as follows: (1) All three types of social capital have a significant promoting effect on the realization of mutual assistance for the elderly in rural areas, among which bridging social capital has the most significant effect; (2) psychological capital plays a partial mediating role in the influence of the three kinds of social capital on the realization of mutual assistance for the elderly in rural areas, and the intermediary role is the strongest in the effect of linking social capital on the realization of mutual assistance for the elderly in rural areas, but the overall effect is not high; (3) among the four dimensions of psychological capital, self-efficacy, optimism, and tenacity all have certain intermediary effects, but the intermediary effect of hope is not significant; and (4) there are significant gender and regional differences in the impact of social capital and psychological capital on the realization of mutual aid for the elderly in rural areas. Linking social capital and psychological capital have a stronger effect on men, while bonding social capital and bridging social capital have a stronger effect on women. Bonding social capital plays a stronger role in promoting the rural elderly in the western region, while linking social capital has more significantly promoted the rural elderly in the central region to participate in mutual assistance for the elderly.

Although it is not uncommon to apply social capital to the study of mutual aid for the elderly in rural China [20,21], this paper extended it to the cooperative study of social capital, psychological capital, and the realization of mutual aid for the elderly in rural China for the first time. In addition, social capital is usually measured by a single indicator and is mainly carried out in cities and developed areas [16,39]. However, this paper provided a more comprehensive measurement by establishing a social capital structure more suitable for its field characteristics in rural China. Consistent with previous research results, social capital from relatives, peers, and the government will significantly promote the rural elderly to participate in the realization of mutual assistance [17]. However, this paper also found that in the three major regions of China, the role of social capital from peers is stronger than that from relatives and the government, which are usually regarded as the main bodies of security supply for the elderly in rural areas. This conclusion makes it particularly necessary to solve the problem of shrinking social scope and decreasing peer interaction faced by the elderly in rural areas and reshape the social network of the elderly in rural areas. At the same time, this paper also extended the research of psychological capital to the field of rural mutual aid pension for the first time in China. The results showed that psychological capital plays a mediating role in the relationship between different types of social capital and the realization of mutual assistance for the elderly in rural areas, which is consistent with the existing research results on psychological capital promoting human behavior [33,34,37]. However, this paper also found that in the rural context of China, support from the government is more important for improving the psychological capital of the rural elderly and further promoting the realization of mutual aid for the elderly. This indicates that the Chinese government needs to improve its policy supply model and take the impact of policies on the psychology of the rural elderly as an important consideration. Another difference from the existing research is that among the four dimensions of traditional psychological capital, hope does not play a good intermediary role. The insight from this is that in the context of carrying out the publicity of the new elderly care model, especially in China’s rural areas where resources are relatively scarce and the actual support level of the elderly care model may be difficult to achieve up to the expected level, cultivating too high a level of hope may not be conducive to the participation of the elderly. In addition, this study found that social capital and psychological capital among the elderly of different genders have different mechanisms for the realization of mutual assistance for the elderly in rural areas. This research result reveals that the intervention focus on promoting the participation of the elderly of different genders in the mutual assistance for elderly care should be different. For elderly males, psychological factors and policy factors have more significant intervention effects. Therefore, the focus should be on the supply and subsidy of policies related to the mutual assistance for elderly care, as well as the impact of policies on the psychology of the elderly. The elderly females should start from the village and family level; focus on cultivating a culture of mutual assistance of good neighborliness, friendship, and mutual help; and further play the role of bonding social capital and bridging social capital.

## Figures and Tables

**Table 1 ijerph-20-00415-t001:** Basic information about the provincial distribution of effective samples.

Item	Eastern Region	Central Region	Western Region	Total
Sample size	737	552	419	1708
Proportion (%)	43.12	32.36	24.52	100

**Table 2 ijerph-20-00415-t002:** Description of core variables.

Variable	Dimension	Item	Assignment
Dependent variable			
The realization of mutual assistance for the elderly in rural areas	Participation of the rural elderly in mutual assistance	Have you ever participated in any of the following types of voluntary services (elderly care/medical care/helping the disabled/helping the poor) in the past year?	Not participated = 1, participated = 2
Independent variable			
Bonding social capital	Scale of social network	The amount of favors you spent on other people’s business (weddings, funerals, etc.) during the previous year?	Take the logarithm of the actual value (except 0).
	Strength of social network	Are you satisfied with your social life?	Score 1~5 = 1, 6~10 = 2
	Environment of social network	Are you satisfied with your family relationship?	Score 1~5 = 1, 6~10 = 2
	Social information exchange	What was your communication expenditure in the previous year?	Take the logarithm of the actual value (except 0).
Bridging social capital	Scale of social network	In the past two years, have you participated in offline activities of interest organizations or non-governmental public welfare organizations, such as clansmen association/township association, alumni association, sports and entertainment?	Not participated = 1, participated = 2
	Strength of social network	What are your gifts and inheritance income from others in 2018?	No relevant income = 1, relevant income = 2
	Environment of social network	What do you think of the trust level between people now?	Score 1~5 = 1, 6~10 = 2
	Social information exchange	Have you ever used the Internet to socialize?	No Internet social = 1, Internet social = 2
Linking social capital	Scale of social network	Are you a member of the Communist Party of China?	Member of the CPC = 2, other political features = 1
	Strength of social network	What do you think of the current social security situation in general?	Score 1~5 = 1, 6~10 = 2
	Environment of social network	How much do you trust the government?	The arithmetic mean of trust degree of township government, local government, and central government, 1~5 = 1, 6~10 = 2
	Social information exchange	Do you browse current political information online?	Never browsed = 1, almost every day/several times a week/several times a month/several times a year = 2
Mediation variable			
Psychological capital	Self-efficacy	Do you have the ability and knowledge to comment on politics?	Full compliance, comparative compliance = 2, less compliance, complete non-compliance = 1
		Are you no longer able to comment on environmental issues?	Full compliance, comparative compliance = 1, less compliance, complete non-compliance = 2
	Optimism	On the whole, are you satisfied with your life?	Score 1~5 = 1, 6~10 = 2
	Hope	If you have a next life, are you still willing to be a Chinese?	Disagree, disagree, disagree = 1, agree, agree = 2
	Tenacity	Do you think every Chinese has the same opportunity to gain wealth and happiness?	Disagree, disagree, disagree = 1, agree, agree = 2
Control variables			
Key influencing factors of mutual assistance for the elderly	Gender	What is your gender?	Gender: male = 1, female = 2
	Age	May I ask your age?	Actual age (years)
	Religion	What is your religion?	No religion = 1, religious = 2
	Degree of education	How much education do you have?	No schooling = 1, primary school, junior high school = 2, high school, technical secondary school, vocational high school = 3, junior college, college degree = 4, master’s degree or above = 5
	Whether there is a spouse	Do you have a spouse?	Unmarried, divorced, widowed, unclear = 1, first marriage with spouse, remarriage with spouse, cohabitation = 2
	Working conditions	What is your work situation?	With work = 2, without work = 1
	Relative economic situation	What level do you think your socio-economic status is in the local area?	Upper = 5, upper middle = 4, middle = 3, lower middle = 2, lower = 1
	Family size	How many people are there in your family?	Specific number (person)
	Number of children	How many children do you have now?	Actual number of children (person)
	Region	What is your area?	Eastern region = 1, central region = 2, western region = 3

**Table 3 ijerph-20-00415-t003:** Descriptive statistical results of variables.

	Minimum	Maximum	Mean Value	Standard Deviation
Realization of mutual assistance for the elderly in rural areas	1.00	2.00	1.08	0.269
Bonding social capital	0.500	3.095	2.140	0.589
Bridging social capital	1.00	2.25	1.266	0.208
Linking social capital	1.000	2.000	1.423	0.208
Psychological capital	1.0	2.0	1.551	0.201
Age	1	2	1.53	0.499
Gender	59	69	64.13	2.999
Religion	1	2	1.12	0.324
Degree of education	1	4	1.84	0.619
Whether there is a spouse	1	2	1.85	0.355
Number of children	0	10	2.38	1.120
Working conditions	1	2	1.64	0.479
Family size	1	20	5.59	2.846
Relative economic situation	1	5	2.40	0.992

**Table 4 ijerph-20-00415-t004:** Correlation test of core variables.

	Realization of Mutual Assistance for the Elderly	Bonding Social Capital	Bridging Social Capital	Linking Social Capital	Psychological Capital
Realization of mutual assistance for the elderly	1	—	—	—	—
Bonding social capital	0.095 ***	1	—	—	—
Bridging social capital	0.163 ***	0.143 ***	1	—	—
Linking social capital	0.143 **	0.168 ***	0.424 ***	1	—
Psychological capital	0.125 ***	0.141 ***	0.173 ***	0.265 ***	1

** At the 0.05 level (double tail), the correlation is significant. *** At the 0.01 level (double tail), the correlation is significant.

**Table 5 ijerph-20-00415-t005:** The impact of bonding social capital, bridging social capital, and linking social capital on the realization of mutual assistance for the elderly in rural areas.

Variable	Model I The Realization of Mutual Assistance for the Elderly in Rural Areas	Model II The Realization of Mutual Assistance for the Elderly in Rural Areas
Age	−0.045 *	−0.035
Gender	−0.247 **	−0.149
Religion	0.364	0.251
Degree of education	0.432 ***	0.145
Whether there is a spouse	−0.217	−0.223
Number of children	−0.195 **	−0.224 **
Family size	0.035	0.032
Working conditions	−0.003	0.015
Relative economic situation	0.272 ***	0.133
Region	0.017	0.175
Bonding social capital		0.413 **
Bridging social capital		1.972 ***
Linking social capital		1.133 **
Bonding social capital^2^		0.104 **
Bridging social capital^2^		0.710 ***
Linking social capital^2^		0.441 ***
Constant term	−0.789	−5.668 **
(Pseudo)R2	0.036	0.082
Overall significance of the model	Yes	Yes

Tips: *** *p* < 0.001, ** *p* < 0.05, * *p* < 0.1.

**Table 6 ijerph-20-00415-t006:** Regression estimation results of bonding social capital, bridging social capital, and linking social capital and their effects on psychological capital.

Variable	Model III Psychological Capital	Model IV Psychological Capital
Age	0.001	0.002
Gender	−0.033 ***	−0.022 ***
Religion	−0.011	−0.014
Degree of education	0.038 ***	0.023 ***
Whether there is a spouse	−0.034	−0.002
Number of children	−0.007 *	−0.010 **
Family size	−0.003	−0.003
Working conditions	0.005	0.002
Relative economic situation	0.048 ***	0.037 ***
Region	−0.001	0.001
Bonding social capital		0.025 ***
Bridging social capital		0.058 **
Linking social capital		0.150 ***
Bonding social capital^2^		0.007 ***
Bridging social capital^2^		0.021 **
Linking social capital^2^		0.055 ***
Constant term	1.389 ***	1.113 ***
R2	0.090	0.129
Overall significance of the model	Yes	Yes

Tips: *** *p* < 0.001, ** *p* < 0.05, * *p* < 0.1.

**Table 7 ijerph-20-00415-t007:** KHB test results of the psychological capital mediation effect.

Model	Model V	Model VI	Model VII
**Independent variable**	Bonding social capital	Bridging social capital	Linking social capital
Total effect	0.590 ***	2.748 ***	2.261 ***
Direct effect	0.521 ***	2.511 ***	1.925 ***
Indirect effect	0.070 *	0.237 ***	0.336 ***
Intermediary effect (%)	11.86	8.62	14.86
R2	0.05	0.08	0.06
Control variable	Yes	Yes	Yes

Tips: *** *p* < 0.001, * *p* < 0.1.

**Table 8 ijerph-20-00415-t008:** KHB test results of the mediating effect of psychological capital on each dimension.

	Model VIII	Model IX	Model X	Model XI	Model XII
Independent variable: social capital					
Intermediate variable	Psychological capital	Self-efficacy	Optimism	Hope	Tenacity
Total effect	2.902 ***	2.873 ***	2.889 ***	2.888 ***	2.890 ***
Direct effect	2.552 ***	2.726 ***	2.775 ***	2.889 ***	2.824 ***
Indirect effect	0.350 ***	0.147 ***	0.113 *	−0.001	0.066 **
Intermediary effect (%)	12.06	5.13	3.91	_	2.28
Control variable	Yes	Yes	Yes	Yes	Yes

Tips: *** *p* < 0.001, ** *p* < 0.05, * *p* < 0.1.

**Table 9 ijerph-20-00415-t009:** Heterogeneity verification results.

Variable	Model XIII	Model XIV	Model XV	Model XVI	Model XVII
	Male	Female	Eastern region	Central region	Western region
Bonding SC	0.113 *	0.729 **	0.403	0.056	1.286 ***
Bridging SC	1.042	3.232 ***	1.755 **	3.118 ***	1.083 *
Linking SC	1.906 ***	−0.511	0.672	1.563 *	0.917
PC	2.298 ***	1.023 *	2.321 ***	−0.287	2.868 ***
CV	YES	YES	YES	YES	YES
Constant term	−7.720 ***	−5.375	−9.810 **	−1.604	−8.677 *
Pseudo R2	0.1155	0.1027	0.1011	0.1683	0.1393

Tips: *** *p* < 0.001, ** *p* < 0.05, * *p* < 0.1.

## Data Availability

The datasets used by the researchers in the analysis of the study are available by the authors upon reasonable request. We confirm that informed consent was obtained from all participants.

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
