# Peer review of "Social Capital and the Realization of Mutual Assistance for the Elderly in Rural Areas—Based on the Intermediary Role of Psychological Capital"

_ijerph, 2022, doi:10.3390/ijerph20010415_

Round 1

Reviewer 1 Report

The author has talked about a very interesting and meaningful topic regarding on the rural elderly assistant in China, and divided them into three types: combination type, bridging type and linking type social capital, this is an interesting research topic, generally speaking, the paper organized well, while I have a lot of comments below:

(1) I have read clearly of the paper, yes, it is a good topic to discuss, The first problem is in the abstract, normally, when you talk about the psychological capital has a partial mediating effect, I want to see the specific value of the impact effect, similarly, the mediating effect which is more significant in bonding and linking social capital, you should tell the author the influence coefficients, which will be more straightforward.

(2) In the introduction, I haven’t seen the big practical meaningful and value for the research question of “social capital and rural mutual assistant”. That is to say, you must link the topic with the rural revitalization strategy in China, also, I haven’t seen the literature summarization for the previous articles, you haven’t evaluated them and the how you find the “research gap”, and following your topic to make up the “gap”.

(3) Some place the expression. Formatting problems, such Siphon effect” etc….,You need to pay attention the formatting.

(4) I have read and learned all of the four hypotheses, the analysis is clear, while I think all the four hypotheses can very easily concluded from the previous literature and the author haven’t put forward some new insights from himself, I think most possibility, the author can do some creative works regarding on the intermediate effects of psychological capital between social capital and rural mutual assistant.

(5) The theoretical framework is quite simply, I think the author can enrich the theoretical framework by clear present the positive/negative of intermediate effect of psychological capital between three types of social capital and mutual elderly assistance. You can also point out which may have the mediating effect, which may not exist mediate effect, in which routes, the mediating effect will be bigger, while will be less.

(6) The data obtained are good, and from a database of CSS. What I am concerning is that the survey from a national comprehensive survey may do service for the national statistic and when you apply into your specific questions, you may need to do “some reasonable dealt” for the independent variables and dependent variables, which makes all the readers convincing.

(7) The author need to make more explanation why the “poverty alleviation” ,i.e help the poor can be the indicate variable for mutual assistant, I mean the poverty alleviation should be a more up-level index system, it has more connotation and meaning, it seems more widely than the rural assistant.

(8) In the control variables section, I think the control variables are less, maybe you should refer to the location characteristics, such as the distance to the central city, also the policy variables, such as does the government promoted some advocating program in the community, etch…..

(9) I think the author should add equation, the linear regression equation between Z to X1,X2,X3, and also explain clear what are the estimated regression parameters of a, b and R.  

(10) I think the author no need to talk too much contents regarding on the influence of control variables on the mutual assistant, just focus on the key variables and the mediating variables and take emphasize on the mechanism analysis and to response your research hypotheses, then to make some detail analysis. The author should marked in model V, VI, VII, with bonding social capital, bridging social capital and connecting social capital, which make it more clear .  The authors have themselves made mistake when using combining social capital replace bonding social capital. The conclusion and policy recommendations are good, but you should also emphasize that the government should try multiple channels to accumulated the psychological capital stocks as you said in the abstract. 

I have very clearly read your paper and give my suggestions for the improvement of your paper, I hope you can absorb these comments and do great improvements for your manuscript.

Author Response

Thank you for your valuable comments on this article. Based on your suggestions, we have made the following changes to the content of this article.

  • I have read clearly of the paper, yes, it is a good topic to discuss, The first problem is in the abstract, normally, when you talk about the psychological capital has a partial mediating effect, I want to see the specific value of the impact effect, similarly, the mediating effect which is more significant in bonding and linking social capital, you should tell the author the influence coefficients, which will be more straightforward.

  1. We completely updated the abstractsection and added the corresponding coefficients after each result (lines 9-43).

  • In the introduction, I haven’t seen the big practical meaningful and value for the research question of “social capital and rural mutual assistant”. That is to say, you must link the topic with the rural revitalization strategy in China, also, I haven’t seen the literature summarization for the previous articles, you haven’t evaluated them and the how you find the “research gap”, and following your topic to make up the “gap”.

  1. We redesigned the introduction part and added the necessity of research on social capital and rural mutual assistance for elderly care. In lines 60-66 and 74-77 of the manuscript, we add the necessity of studying social capital and mutual support for old-age care. In addition, in lines 96-103 of the manuscript, this paper adds an explanation on the necessity of social capital research in rural China. This paper adds the existing research results from the aspects of the research source of social capital and the role of social capital in a larger scope, and on this basis further adds the research gapand the contribution of this paper.

  • Some place the expression. Formatting problems, such Siphon effect” etc….,You need to pay attention the formatting.

  1. We have corrected the non-standard expression in the article as much as possible. At the same time, as for the language of the manuscript, we asked a doctor of our university who is a native English speaker to polish the manuscript. If the article has further language problems, we will use the editing service provided by the editorial office to make further changes.

  • I have read and learned all of the four hypotheses, the analysis is clear, while I think all the four hypotheses can very easily concluded from the previous literature and the author haven’t put forward some new insights from himself, I think most possibility, the author can do some creative works regarding on the intermediate effects of psychological capital between social capital and rural mutual assistant.

  1. We choose to add the test of different dimensions of psychological capital between social capital and the realization of mutual support for the elderly in rural areas, and form some new conclusions based on the empirical results (lines 461-496).

  • The theoretical framework is quite simply, I think the author can enrich the theoretical framework by clear present the positive/negative of intermediate effect of psychological capital between three types of social capital and mutual elderly assistance. You can also point out which may have the mediating effect, which may not exist mediate effect, in which routes, the mediating effect will be bigger, while will be less.

  1. In lines 270-272 of the manuscript, we modified the research framework of this paper to make it fit with the revised research content.Thank you for your helpful suggestions.

  • The data obtained are good, and from a database of CSS. What I am concerning is that the survey from a national comprehensive survey may do service for the national statistic and when you apply into your specific questions, you may need to do “some reasonable dealt” for the independent variables and dependent variables, which makes all the readers convincing.

  1. In order to solve this problem, first of all, based on the specific problems of the research, this paper operationalizes the items, and resolves some data inaccuracies to a certain extent by means of 1/2 assignment (lines 326-327). In this revision, the square of the social capital variable is constructed to ensure the robustness of the research conclusion (lines 399-401).

  • The author need to make more explanation why the “poverty alleviation” ,i.e help the poor can be the indicate variable for mutual assistant, I mean the poverty alleviation should be a more up-level index system, it has more connotation and meaning, it seems more widely than the rural assistant.

  1. This is a translation error. We modify the translation of this item to “help the poor ”(lines 326-327) in order to reduce its scope of meaning to some extent. In fact, under China divides the poor population by family unit and the standard definition of poverty, the poor are concentrated in the rural elderly who live alone, have no children or split their families from their children. However, the support level provided by the rural elderly themselves is relatively limited, so their actions to help the poor are more concentrated on other elderly people in the same village. This is the reason why this item is included in this paper.

  • In the control variables section, I think the control variables are less, maybe you should refer to the location characteristics, such as the distance to the central city, also the policy variables, such as does the government promoted some advocating program in the community, etch…..

  1. According to your suggestion to expand the scope of control variables, we add two control variables, religious belief and education level, in view of the items contained in the open database (lines 326-327). The results show that when the above control variables are included, the significance of the dependent variable does not change, but the β value decreases slightly.

  • I think the author should add equation, the linear regression equation between Z to X1,X2,X3, and also explain clear what are the estimated regression parameters of a, b and R.

  1. This paper adds the equation of independent variable versus intermediary variable Z (line 334). At the same time, the equation of the KHB model construction part is modified to make the basic idea of the khb model more obvious (lines 352-364).

(10) I think the author no need to talk too much contents regarding on the influence of control variables on the mutual assistant, just focus on the key variables and the mediating variables and take emphasize on the mechanism analysis and to response your research hypotheses, then to make some detail analysis. The author should marked in model V, VI, VII, with bonding social capital, bridging social capital and connecting social capital, which make it more clear .  The authors have themselves made mistake when using combining social capital replace bonding social capital. The conclusion and policy recommendations are good, but you should also emphasize that the government should try multiple channels to accumulated the psychological capital stocks as you said in the abstract.

10.Based on your review comments,We have deleted the discussion of control variables in the original manuscript (line 405). The study focuses on the discussion of the direct effect and the mediation effect. Further, corresponding dependent variables were added to models V,VI and VII to make the model clearer (lines 459-460). We are very sorry for some low-level errors in the manuscript. We have modified the wrong use of COMBING and BONDING (lines 297,303,319,456). We also revised our findings as a whole. In this part, we first mention the research contribution of this paper, and then based on the research conclusion, through comparison with existing research, find out the parts that are different from the previous research conclusions, and put forward suggestions based on this, so that the research conclusion is not limited to China, but has a stronger reference value (lines 505-536).

Thank you again for your helpful revision comments. We believe this is a great help to further improve our research

Reviewer 2 Report

Dear Authors, all comments and directions in the attachment. 

Author Response

Please refer to the attachment for the reply results of the review comments.

Round 2

Reviewer 1 Report

I have read clearly of the revised manuscript, I can see the author has done great efforts for the improvement of the paper, yes this is deserving to be admired: while, I have some little comments to help you improve the paper further.

(1) In the introduction, add the partial mediating coefficients is enough, no need to add some other relevant information, P < 0.01; and so on.

(2) In the introduction section, more sentences are need to added to introduce the research problem step by step: Rural revitalization is important for Chinese government——rural governance is an important part for rural revitalization, the village elderly governance is important——the research focus on rural elderly assistance is enough, while from the perspective of social capital is lacking ——Our contribution is not only reveal the direct and also the indirect(mediating) influence mechanism between them——This research my do service for the rural elderly assistant and helps to the local governments to take targeted measures.

(3) The response for my comments (4) is insufficient, I don’t think the modification for the research hypotheses have made improvements, if you haven’t new findings and hypotheses for the research design, if it is possible to make some heterogenous test and further analysis. In addition, you say the change in (lines 461-496). I guess you have made a mistake.

(4) The authors haven’t done some improvement and revision as my comments for the research theoretical framework, what is the direct impacts of SC on rural elderly assistance, PC has done what mediating effects Between SC and elderly assistance, these cannot be clearly find in the framework.

(5) The sentence organizations in conclusion section are confusing, why the research results appear at the last of the paper, you should present the results and then give the policy recommendation, you should adjust the orders of the sentences.

My suggestions are helping you to develop your paper, so no need to hurry, a good paper can help to make meaningful research, and of course can read and cited by more following scholars.

Author Response

请参阅附件。

Reviewer 2 Report

Dear Authors,

Thank you for including the important suggestions in the revision of the manuscript and the responses to the report.

After the reconstruction of the manuscript, mainly Part: Introduction, the manuscript is more understandable for readers. General the manuscript correction is satisfying, but there are some advices extra:

Based on the comparison the both versions of the manuscripts I noticed:

1. Some parts of the article have been changed, as well as the values in Table 5 and Table 6; some variables (Level of education) has been added and highlighted in red, so is it the result of this correction in model? 

2. Some doubts with Figure 1. - please add the explanation below (f.e. own elaboration?? or the appropriate note).

3. The abstract is clear but very detailed, in my opinion in the section: Results - the extra details, such as numbers (f.e. β= 0.413, P < 0.05;), can be omitted - put only main results. All details are include in main text in Part 4, so it is not demanded to put so extra details in Abstract. Please, for the consideration.

4. In my opinion, the Part: discussion should a bit deeply refers to the results of the research the others authors in describing problem.

5. Please, delete minor text editing errors. 
